# Yeast Microbiota during Sauerkraut Fermentation and Its Characteristics

**DOI:** 10.3390/ijms21249699

**Published:** 2020-12-18

**Authors:** Paweł Satora, Magdalena Skotniczny, Szymon Strnad, Katarína Ženišová

**Affiliations:** 1Department of Fermentation Technology and Microbiology, Faculty of Food Technology, University of Agriculture in Krakow, Balicka 122, 30-149 Krakow, Poland; magdalena.skotniczny@urk.edu.pl (M.S.); szymon.strnad@urk.edu.pl (S.S.); 2Department of Microbiology, Molecular Biology and Biotechnology, National Agricultural and Food Centre, Food Research Institute, Priemyselna 4, P.O. Box 25, 824 75 Bratislava, Slovakia; katarina.zenisova@nppc.sk

**Keywords:** sauerkraut, yeast microbiota, 5.8S-ITS rRNA gene, *Debaryomyces*, resistance for stress conditions, volatile profile

## Abstract

Sauerkraut is the most important fermented vegetable obtained in Europe. It is produced traditionally by spontaneous fermentation of cabbage. The aim of this study was to determine biodiversity of yeasts present during fermentation of eight varieties of cabbages (Ambrosia, Avak, Cabton, Galaxy, Jaguar, Kamienna Głowa, Manama and Ramco), as well as characterize obtained yeast isolates. WL Nutrient Agar with Chloramphenicol was used to enumerate yeast. Isolates were differentiated using RAPD-PCR and identified by sequencing of the 5.8S-ITS rRNA gene region. The volatiles production was analyzed using SPME-GC-TOFMS. Our research confirmed that during sauerkraut fermentation there is an active growth of the yeasts, which begins in the first phases. The maximal number of yeast cells from 1.82 to 4.46 log CFU g^−1^ occurred after 24 h of fermentation, then decrease in yeast counts was found in all samples. Among the isolates dominated the cultures *Debaryomyces hansenii*, *Clavispora lusitaniae* and *Rhodotorula mucilaginosa*. All isolates could grow at NaCl concentrations higher than 5%, were relatively resistant to low pH and the presence of lactic acid, and most of them were characterized by killer toxins activity. The highest concentration of volatiles (mainly esters and alcohols) were produced by *Pichia fermentans* and *D. hansenii* strains.

## 1. Introduction

Poland is the fourth largest producer of sauerkraut in Europe, and its annual consumption in our country is about 3 kg per capita [1]. This fermented vegetable is produced by spontaneous fermentation of cabbage leaves, mainly by lactic acid bacteria [2]. As a result of the fermentation process, the fermenting sugars found in the vegetable are transformed into lactic and acetic acid, ethanol, CO_2_, mannitol and other compounds [3].

The quality of the product depends mainly on the microbiota contained on the crude vegetable [4]. The development of microorganisms in the appropriate sequence is necessary to obtain the taste and aroma characteristic of sauerkraut [5].

In the initial phase of fermentation of vegetable, among the large variety of microorganism, yeasts represent a large group. Sometimes it even happens that they begin to dominate and control the fermentation process. In the fermentation of vegetable silages, the negative activity of yeast leads to an increase in pH, which can be caused by one of two processes. The first is related to the fact that the yeasts present in the silage carry out ethanol fermentation using sugars, which should be fermented by the bacteria to lactic acid. The second process is related to the development of *Geotrichum candidum* and *Candida mycoderma* yeasts on the silage surface under aerobic conditions. The above microorganisms use organic acids stabilizing the product as a carbon source. This reduces the acidity, increases the pH and creates conditions for the development of spoilage bacteria. Yeast is also responsible for the production of a large amount of gases [6]. Too soft silage structure may be associated with insufficient salt concentration. It should be added to the freshly shredded cabbage in such quantity as to stimulate the growth of the desired lactic bacteria in the right order. Lactic acid and salt determine the correct hardness of the product by inhibiting pectinolytic enzymes [7]. Very large economic losses in the industrial production of sauerkraut causes the appearance of a defective pink color associated with the development of yeast *Rhodotorula*. They produce carotenoids, which give the silage pink color. This disadvantage is usually associated with too high concentration of salt added, which exceeds 3% [8].

Disadvantages in the form of strange aftertaste and aroma, which are unspecific for pickled products and are associated more with the yeast industry, are most often caused by a too fast fermentation process carried out at high temperature. These conditions favor the growth of aerobic microorganisms, mainly yeasts and molds, which produce metabolites responsible for undesirable taste and smell. Sauerkraut in which yeast and mold develops is characterized by a high content of esters responsible for the aroma of raw cabbage, while the amount of lactic acid is much lower in it than in products in which the contaminating microbiota did not occur [7].

Most of the so far published articles related to the microbiology of the sauerkraut concern LAB (Lactic Acid Bacteria). There is very little information regarding the presence of yeast during fermentation of sauerkraut and their characteristics. The aim of this study was to determine biodiversity of yeasts present during fermentation of different varieties of cabbages as well as characterize obtained yeast isolates. For the experiments, eight varieties of cabbages, most often used by Polish farmers for sauerkraut production were used.

## 2. Results

### 2.1. Yeast Population Kinetics

The isolation was done in 0., 1., 2., 3., 7., 10. and 14. day of fermentation (Figure 1). Quantitative analysis showed differences in total yeast contents during fermentation of cabbages of different varieties. The fresh cabbages contained from 0.60 (Cabton) to 3.74 (Manama) log CFU of yeasts per gram. For the majority of trials in the following day, there was an increase in the number of yeast cells (Manama, Ramco, Ambrosia, Jaguar, Cabton), reaching their maximum. In the sauerkraut of the Kamienna Głowa, Galaxy and Avak varieties, from the very beginning of fermentation, a gradual decrease in the number of yeast cells has been reported.

After 24 h of fermentation, the decrease in yeast counts was found in all samples. After two days of the process in Cabton cabbage (with the smallest initial number of yeast cells), the presence of these microorganisms was not observed; a similar tendency was observed in the majority of the other samples after 72 h of the process. The exceptions were the Kamienna Głowa and Ambrosia varieties, in which on the third day of fermentation, 2.28 and 0.78 log CFU g^−1^ were still detected. During sampling in 7th day of fermentation no yeast was found.

### 2.2. Biodiversity of Yeasts during Fermentation

A total of 246 pure yeast cultures were isolated from various stages of fermented sauerkraut from eight varieties of cabbages.

The RAPD-PCR method was used to differentiate isolates and reduce their number before further analysis. Isolated yeasts were classified into 25 groups based on distinct electrophoretic patterns (Figure 2). Representatives of each groups of RAPD patterns were analyzed by 5.8S-ITS (internal transcribed spacer) PCR-RFLP (polymerase chain reaction-restriction fragment length polymorphism) (Table 1) and were identified by 5.8S-ITS rRNA gene region sequencing.

The RFLP analysis showed that the isolates belong to nine yeast species. This was confirmed by sequencing, with 25 electrophoretic patterns obtained by RAPD (Figure 2), classified into 17 different strains by comparing obtained sequences with those available in the GenBank NCBI database, considering an identity threshold of at least 98% (Table 1).

Among the isolates dominated the cultures *Debaryomyces hansenii* and *Clavispora lusitaniae*—four strains each, and *Rhodotorula mucilaginosa*—three strains. The presence of single strains of *Cryptococcus macerans*, *Nakazawaea holstii*, *Meyerozyma guilliermondii*, *Candida sake*, *Pichia fermentans* and *Tausonia pullulans* were also found.

Figure 3 present participation of individual yeast strains identified during spontaneous fermentation of different cabbage varieties at various fermentation stages.

In the case of 4 analyzed varieties (Galaxy, Kamienna Głowa, Manama, Ambrosia) a large diversity of the yeast population was found, in the others much smaller. The yeast population in terms of species and strains was the most diverse at the beginning of fermentation, in the course of time, selected cultures remained.

In the prepared for fermentation cabbage of the Galaxy variety, *C. lusitaniae* X, XI and *D. hansenii* XII strains prevailed. A similar composition of yeast microbiota was found on the cabbage of the Jaguar variety. Kamienna Głowa and Manama samples were similar and contained cultures of *Cry. macerans* I and *R. mucilaginosa* III. In addition, in the first case *N. holstii* II was detected, and the second —*D. hansenii* XII and *R. mucilaginosa* VIII. The Ambrosia cabbage was characterized by the most diverse yeast population—*D. hansenii* (strains IV, XII), *C. lusitaniae* (XIII), *P. fermentans* (XIV) as well as *T. pullulans* (XVII) were identified. Similarly, in the structure of yeast microbiota, Ramco was distinguished, on the surface of which *M. guilliermondii* VI and *C. sake* VII were found. Cabbage of Avak cultivars were the least diverse in this respect, in which only yeast *D. hansenii* XII was isolated.

After 24 h of fermentation the growth of *Basidiomycota* yeast was significantly reduced and cultures with fermentative metabolism, such as *D. hansenii* (strains IV, XII, XV, XVI), *C. lusitaniae* (XI, XIII) and *P. fermentans* (XIV) started to prevail.

In subsequent days of the process, at the time of a significant decrease in the amount of yeast (Figure 1), the strain *D. hansenii* XII began to dominate, regardless of the variety of the cabbage subjected to fermentation.

### 2.3. Production of Volatile Components by Isolates

A total of 31 different aroma components were detected in the samples (Figure 4). Alcohols and esters constituted the most numerous groups. Based on the cluster analysis and heat map, the strains analyzed were divided into three groups. The first was cultures forming small amounts of volatile compounds, such as *R. mucilaginosa* (strains III, VIII, IX), *Cry. macerans* (I), *N. holstii* (II), *C. lusitaniae* (V, XIII), *C. sake* (VII) and *T. pullulans* (XVII), the second strains producing average amounts of volatile compounds—*C. lusitaniae* (X, XI), *M. guillermondii* (VI) and *D. hansenii* (IV), and to the third largest producer of volatile compounds—*D. hansenii* (XII, XV, XVI) and *P. fermentans* (XIV). The most volatile components were formed by *P. fermentans* XIV—producing almost all analyzed components, some of them such as ethyl acetate were produced in more than 10 times larger quantities than in the case of other isolates. The presence of high concentrations of other esters, such as isobutyl acetate, isoamyl acetate, 2-phenylethyl acetate, ethyl dodecanoate and others were also characteristic of these samples. Several other volatile compounds were also unique to selected cultures, e.g., ethyl 2-hydroxypropanoate was only present in the samples of *M. guillermondii* VI, cyclohexanol—*C. lusitaniae* XI, or cyclohexanone—*C. lusitaniae* V and XI. Isolates producing smaller amounts of volatile components mainly produced fewer volatile substances, such as fatty acid esters, alcohols with more than eight carbon atoms per molecule.

### 2.4. Resistance of Isolates to Selected Stress Factors Present during Sauerkraut Fermentation and Killer Activity

Most of the tested isolates can be classified as halophiles, because they grew very well even in an environment containing 10% sodium chloride. *N. holstii* II, *C. lusitaniae* X, *P. fermentans* XIV and *D. hansenii* XVI strains were an exception and their growth were visibly inhibited on media with 6% NaCl and more (Table 2).

Differences in salt tolerance were also found within a species, e.g., in the case of *D. hansenii* XVI and *C. lusitaniae* X strains, proliferation was significantly inhibited with the addition of 6% NaCl, while the other strains analyzed showed good growth even at 10% salt concentration (Table 2).

Based on the results obtained (Table 2), it was shown that 10 g L^−1^ lactic acid inhibits the growth of almost all tested yeasts. In the case of the *N. holstii* II strain, no growth was observed with the addition of 6 g L^−1^ acid, whereas the strains resistant to high acid concentration were strains *R. mucilaginosa* III, *C. lusitaniae* V, *M. guilliermondii* VI, *C. sake* VII and *D. hansenii* IV growing at a concentration of up to 10 g L^−1^ lactic acid.

Decreasing the pH of the environment to 3.4 significantly inhibited the growth of most yeasts, which explains why, at the final stage of sauerkraut fermentation, these microorganisms are practically absent. Only in case of *M. guilliermondii* VI, *C. sake* VII, *D. hansenii* XV and XVI strains, weak colony growth was noted at pH 3.4 (Table 2).

Eleven of the seventeen analyzed strains showed killer activity (Table 2). The highest killer activity was observed in the yeasts *D. hansenii* XII, *C. lusitaniae* V and *C. sake* VII. All analyzed *D. hansenii* strains were able to inhibit the growth of a sensitive culture. The highest differentiation in this trait was found in the case of the yeast *C. lusitaniae*, among which there were both cultures with strong killer activity, such as strains V and X, and with low or no activity—XI and XIII. The strain with the highest killer activity—*D. hansenii* XII, was predominant during sauerkraut fermentation and was found in the fermenting sauerkraut samples of each variety (Figure 3).

## 3. Discussion

Previous studies of sauerkraut fermentation process included mainly the determination of the quantitative and qualitative composition of bacterial microbiota [5,9,10,11,12], content and formation of selected bioactive components (such as biogenic amines, glucosinolates) [13,14,15,16], the influence of selected physicochemical factors on the above [10,17], as well as management of post-fermentation brine as a waste material for the production of components of biotechnological importance [18,19]. Very little attention was paid to the presence of yeast during this process, and available publications come from many years ago, or contain only mention of yeasts as spoilage organisms [6,8].

In this work we have proved that with properly conducted fermentation process, with anaerobic conditions, yeast growth is typical for sauerkraut fermentation and occurs at the initial period of fermentation (Figure 1). Their high number may affect the sensory characteristics of the product being produced without deterioration in its quality. It is dependent on the salt concentration in brine, as well as the use of the starter culture [10]. In our research, we used a 2.5% NaCl solution, used by Polish producers of sauerkraut, which creates better conditions for the process and more strongly limits the growth of undesirable microbiota than the concentrations used in other regions of the world [10].

Cabbage cultivar strongly influenced the quantitative and qualitative composition of yeast microbiota at the beginning and during its fermentation (Figure 1 and Figure 4). We have not shown a relationship between the content of sugars and dry matter in the cabbage [20] and the amount of yeast. Although the Manama cabbage contained the most sugars (21.7 g kg^−1^ of fructose and 25.8 g kg^−1^ glucose) and the highest number of yeast cells was found on it, a similar trend was not confirmed in the other analyzed samples. Therefore, it should be assumed that other factors, i.e., agrotechnical treatments used during cultivation, as well as climatic conditions present during its development (late or mid-late varieties with a similar harvest time, intended for processing, were used for experiments) could determine the population of yeasts. Similar factors affect the composition of microbiota during spontaneous fermentation of other vegetables/fruits [21,22].

The sauerkrauts were dominated by *D. hansenii* strains (Table 1, Figure 3). Using the RAPD-PCR method and the sequencing of 5.8S-ITS rRNA gene region, four different strains of this species were found, with strain XII occurring in all samples, IV in the Kamienna Głowa and Ambrosia cabbages, and the other two were specific to the Ambrosia variety (Figure 3). Isolated strains significantly differed in terms of sensitivity to stress factors occurring during sauerkraut fermentation, killer activity but also in terms of volatile compounds formed (Table 2, Figure 4). Culture more resistant such as strain IV produced less volatiles. Strain XII was present in most sauerkraut samples (regardless of the variety of cabbage used), formed relatively large amounts of alcohols and esters; it can be assumed that its positive participation in sensory features of sauerkraut could be significant. *D. hansenii* is a halo-, osmo-, and xerotolerant species of yeast. It occurs in many habitats with low water activity, it was also isolated from wine, beer, fruit, cheese, meat and soil as well as from high-sugar products [23]. The generation of volatile and aroma compounds by *D. hansenii* is considered an important contribution to the ripening process of cheeses and fermented sausages [24,25,26]. *D. hansenii* can prevent the formation of lipid oxidation products in fermented sausages, and contributes to improve sensory components, principally flavors, such as ethyl esters [26]. Unfortunately, this yeast can also spoil brine-preserved foods, such as gherkins [23]. Deak and Beuchat [27] report that yeast *D. hansenii* develops only at the beginning of the fermentation process of cabbage and their population gradually dies off. Not without significance is the induction of fermentation properties of *D. hansenii* in the presence of salt [28], which can increase the survival of cultures under anaerobic conditions and stimulate the cells to form larger amounts of volatile compounds positively affecting the silage aroma. However, confirmation of the positive impact of *D. hansenii* cultures on the process of sauerkraut fermentation still requires further research.

The second most common yeast species detected in our analyses was *C. lusitaniae*. This is the first report regarding the presence of these yeast in fermented vegetable. It is a saprobial, fermentative yeast, that has been associated with clinical specimens from immunocompromised patients and is now recognized as an opportunistically infectious organism. It has also been isolated from food and food production processes, including fermentation of agave must and sap, Lager brewing process in Thailand [29], traditional Egyptian dairy products [30], dates, Kopanisti cheese ripening [31] and others. This microorganism is most often found with *D. hansenii* and lactic acid bacteria, but there is no references on the properties of cultures, as well as resistance to stress factors occurring during sauerkraut fermentation, such as the presence of NaCl, lactic acid, low pH, etc. Our research is also the first to determine the ability of these yeasts to show killer activity and produce volatile compounds. The analyzed isolates significantly differed in this respect; they were classified into groups of microorganisms producing small or medium amounts of volatile compounds, forming specific components not obtained from other cultures.

Other yeasts identified during sauerkraut fermentation are also detected in various food products and during their production. *P. fermentans* is responsible for the formation of the cocoa beans’ aroma [32]. Isolated in our research from sauerkraut fermentation strain of *P. fermentans* were the strongest producer of volatiles, especially esters. *Meyerozyma caribbica* (closely related to *M. guilliermondii*) and *N. holstii* are often found together, for example they are the predominant yeasts present in the crushed olives, olive pomace, and fresh table olive [33]. *Cry. macerans* was found in sauerkraut obtained on farms in the south of Poland [34].

Some of the isolates obtained in this work may have interesting properties which can positively shape the quality of food. *D. hansenii* can produce volatile compounds, such as alcohols, esters, carbonyl and sulphur compounds [35]. Furthermore, recent reports on the potential probiotic properties of *Candida lusitaniae* (anamorph *Clavispora lusitaniae*) and *Meyerozyma caribbica* cultures seem very interesting [36]. The finding of such abilities in yeast isolates from sauerkraut could confirm that vegetable silages (also containing numerous LAB cultures) can be a rich source of positively acting microbiota on our organism.

Our preliminary research of commercial and farm-made sauerkraut showed the largest amount of yeast was found in the sauerkrauts produced in the farms located in the Muszyna commune (3.3–4.2 log CFU g^−1^) and in one commercial product (3.4 log CFU g^−1^). In other commercial sauerkraut products analyzed, no yeast was found. Representatives of two species: *Cry. macerans* and *D. hansenii* (3 different strains) predominated among the isolates [34]. These cultures correspond to the strains *Cry. macerans* I and *D. hansenii* IV, XII and XV isolated in this work (Table 1). Their presence in the finished product indicates that single yeast cells can survive the fermentation process, being resistant to the stress factors prevailing in it, such as the presence of salt, anaerobic conditions, increasing lactic acid concentration and decreasing pH. This fact has been confirmed in our research (Table 2). Initially, osmotic pressure associated with the presence of salt and rapid depletion of oxygen in the environment affect microorganisms derived from the raw material. Among the isolates, we only detected cultures that all can grow at NaCl concentrations higher than 5%. Aerobic cultures, such as *Cry. macerans*, *R. mucilaginosa* i *T. pullulans* [32], occurred only in the initial period of fermentation, then they were replaced by yeast with fermentative metabolism (Figure 3). Strains resistant to low pH and the presence of lactic acid produced by lactic acid bacteria were the longest present. Lactic acid has been found to have a weaker inhibitory effect on yeast cells than inorganic acids (8 g L^−1^ lactic acid corresponds to pH 3.4). Earlier studies Hassan et al. [37], showed a similar phenomenon, the effect of eight organic acids (among others acetic and lactic acids) as antifungal agents on the growth of four fungal species (*Aspergillus flavus*, *Penicillium purpurogenum*, *Rhizopus nigricans* and *Fusarium oxysporum*) was different. It has been found that, there is no relationship between the efficiency of organic acid and its final pH.

Another factor influencing the presence of yeasts during fermentation of sauerkraut was their ability to produce killer toxins. Most of the analyzed isolates showed killer phenomena. These antimicrobial substances can inhibit growth of different genera of yeast and play an important role in colonization of the environment [38]. *D. hansenii* has been reported to produce strong and active toxic proteins or glycoproteins, as killer toxins. The research showed that the optimum inhibitory effect of killer toxin was in the presence of NaCl in the environment [39]. The above dependence may explain why during the sauerkraut fermentation, regardless of the cabbage variety used, as the fermentation progressed, the strain of *D. hansenii* XII characterized by the highest killer activity, dominated. A similar phenomenon could be associated with other killer cultures, confirming that in addition to stress factors such as NaCl concentration, lack of oxygen, low pH, and killer activity affects the growth of yeast during fermentation of sauerkraut.

Summarizing, our research confirmed that during properly conducted fermentation process of sauerkraut, with anaerobic conditions, yeast growth occurs always and are present in the initial period of fermentation. We found a large similarity in the composition of yeast microbiota between samples regardless of the cabbage variety used. Among the isolates, we only detected cultures that all can grow at NaCl concentrations higher than 5%, and relatively resistant to low pH and the presence of lactic acid. Most of the cultures were characterized by killer toxins activity. Yeast *D. hansenii* and *C. lusitaniae* predominated in the analyzed samples, because of their beneficial properties may directly affect the sensory characteristics of the finished product, but also increase its pro-health value. However, further research is required to determine the impact of individual cultures on the quality of sauerkraut.

## 4. Materials and Methods 

### 4.1. Cabbages and Fermentation of Sauerkraut

Eight cultivars of cabbage (*Brassica oleracea* var. *capitata* f. *alba*)—Ambrosia, Avak, Cabton, Galaxy, Jaguar, Kamienna Głowa, Manama and Ramco—from planters of southern Poland (harvested from August to October 2016) were used in this study. The obtained cabbage was kept for 24 h before fermentation in room temperature, in dark. After preliminary cleansing from dirt, the core as well as outer leaves were removed. Then, the cabbage was shredded with a sterile slicer MA-GO 612 P to obtain shavings of thickness of 3.5 mm and 4 kg of shredded cabbage was placed in layers a 5-L jar and 2.5% NaCl (*w*/*w*) was added. The experiment was done in triplicate.

### 4.2. Yeast Enumeration and Isolation

Fermentation was carried out in the temperature of 20 °C for two weeks. During the process (0., 1., 2., 3., 7., 10. and 14. day of fermentation), 5 g of sauerkraut and 5 mL of brine were collected from the middle of the jar. Samples were placed in sterile stomacher bags (BagPage 400 mL; Interscience, Woburn, MA, USA) and homogenized for 5 min. Ringer’s solution (sodium chloride 2.25 g L^−1^, anhydrous calcium chloride 0.12 g L^−1^, sodium bicarbonate 0.05 g L^−1^; POCH S.A., Gliwice, Poland) was used for serial decimal dilutions. The appropriate dilutions were plated in triplicate on Petri dishes and poured with WL Agar (Wallerstein Laboratory) (BIOCORP, Warsaw, Poland) supplemented with 100 mg L^−1^ of chloramphenicol.

After the incubation at 20 °C for 5 days (WL Agar) the colonies were enumerated, representative 10 cultures were isolated from each cabbage variety from subsequent fermentation days, based on their colony morphologies (size, shape, color), pure cultures were obtained by streaking on Sabouraud Dextrose with Chloramphenicol LAB-AGAR (BIOCORP, Warsaw, Poland) and identified.

### 4.3. DNA Extraction and RAPD-PCR Analysis

The total yeast genomic DNA was extracted from isolates using a commercial kit, Yeast Genomic Mini AX Spin (A&A Biotechnology, Gdynia, Poland), following the manufacturer’s instructions.

The RAPD-PCR analysis was done according to the method described by Cioch-Skoneczny et al. [22].

### 4.4. PCR-RFLP Analysis and 5.8 S-ITS rRNA Gene Region Sequencing

ITS1 (5′ TCCGTAGGTGAACCTGCGG-3′) and ITS4 (5′-TCCTCCGCTTATTGATATGC-3′) primers were used for the 5.8S-ITS rRNA gene region amplification. The PCR-RFLP analysis and 5.8 S-ITS rRNA gene region sequencing were conducted according to the method described by Cioch-Skoneczny et al. [22]. Sequences were deposited in the GenBank NCBI database with the accession numbers: MK312605-312621.

### 4.5. Production of Volatile Components by Isolates (SPME-GC-TOFMS)

Identified yeast isolates were growing overnight in Sabouraud Dextrose Broth (BIOCORP, Warsaw, Poland). The cells were then centrifuged (735× *g*), resuspended in Ringer’s solution and 10^6^ cells mL^−1^ were inoculated into YNB solution (Yeast Nitrogen Base; Sigma-Aldrich, St. Louis, MO, USA) with 0.5% of glucose and 0.5% of fructose as a carbon source. After 10 days of incubation (25 °C), the cells were removed by centrifugation (735× *g*), and the supernatants were analyzed by SPME-GC-TOFMS, according to the method described by Zdaniewicz et al. [40]. Detected volatiles were identified using the NIST database (http://webbook.nist.gov/chemistry/) and determined semi-quantitatively (μg L^−1^) by measuring the relative peak area of each identified component, in relation to that of the internal standards (4-methyl-2-pentanol and ethyl nonanoate). All tests were carried out three times.

### 4.6. Resistance of Isolates to Selected Stress Factors Present during Sauerkraut Fermentation

Sterile Petri dishes with Sabouraud Dextrose LAB-Agar medium, supplemented with NaCl (2.5, 5, 6, 8 or 10%), lactic acid (6, 8 or 10 g L^−1^) or hydrochloric acid (pH 3.2, 3.4 or 3.6) were prepared. The control was a pH 5.6 medium (without adjusted pH value), with 2.5% NaCl. The plates were divided into 4 parts, and then in each quarter, in five repetitions, 30 μL of yeast suspension in 0.85% NaCl (8 log cells mL^−1^, growing over-night) was placed. The cultures were kept at a temperature of about 20 °C for 48 h. The diameter of the grown colonies was then measured.

### 4.7. Killer Activity

Killer activity was assayed using seeded-agar-plate technique. Killer sensitive strain of *Saccharomyces cerevisiae* (DBPVG 6500) was suspended in Ringer’s solution (~5 log cells mL^−1^) and inoculated into YEPD-MB agar [38]. Then, wells (5 mm) were sterilely cut in the YEPD-MB agar and the potential killer strains were seeded in the wells at 100 μL of yeast inoculum per well. The plates were incubated at 20 °C for up to 7 days. If the tested strain was surrounded by a zone of inhibition fringed with blue color, it was recorded as killer. Killer activity was measured by subtracting diameter of the well from diameter of the inhibition zone.

### 4.8. Statistical Analysis

A heat map and cluster analysis were prepared using the statistical package SPSS 18.0 (SPSS Inc., Chicago, IL, USA).

## Figures and Tables

**Figure 1 ijms-21-09699-f001:**
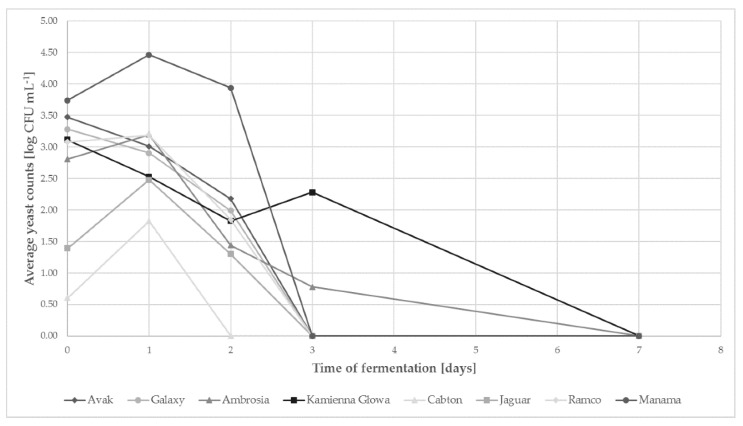
The amount of yeast cells during sauerkraut fermentation of different varieties (standard deviations do not exceed 5%).

**Figure 2 ijms-21-09699-f002:**
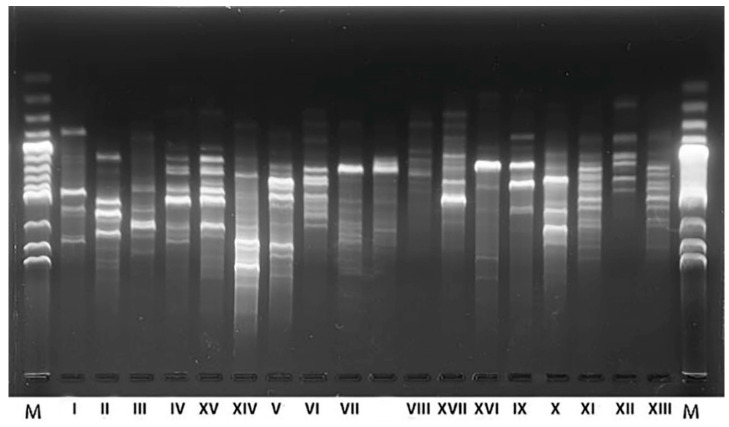
Results of electrophoresis of RAPD-PCR products with M13 starter of yeast isolates from fermenting sauerkraut.

**Figure 3 ijms-21-09699-f003:**
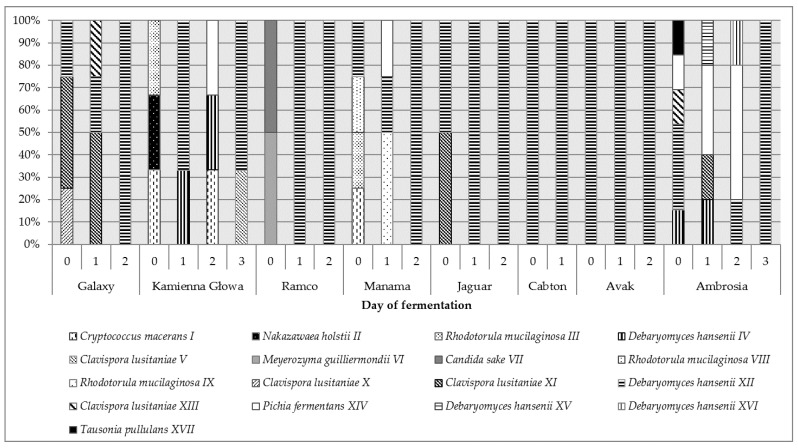
Distribution of yeast strains (%) isolated from the initial stage of different cultivars of cabbage fermentation.

**Figure 4 ijms-21-09699-f004:**
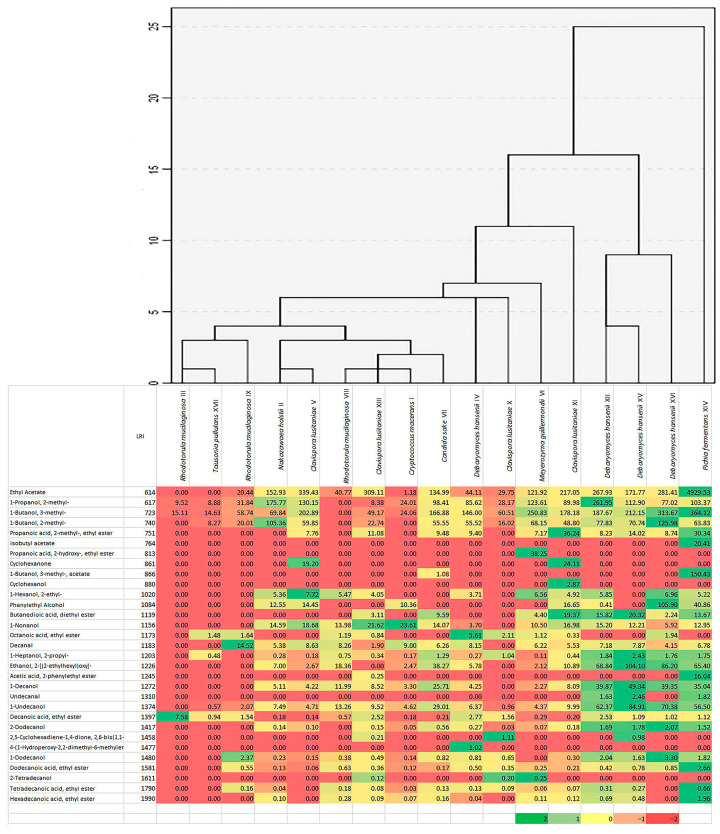
A heat map and cluster analysis results of 31 volatiles [μg L^−1^] produced by yeast isolates from sauerkraut fermentation. The highest content is in the darkest green and the lowest content is in the darkest red.

**Table 1 ijms-21-09699-t001:** Identified species of yeast based on the results of the RFLP analysis and 5.8S ITS rRNA gene sequencing.

Strain	5.8S-ITS rRNA Gene [bp]	Restriction Fragments [bp]			Species Identification by 5.8S‑ITS rRNA Gene Sequencing	Restriction Pattern	Accession No.
		*Cfo*I	*Hinf*I	*Hae*III			
I	650	260 + 200 + 95 + 95	270 + 200 + 175	440 + 80 + 70 + 60	*Cryptococcus macerans*	1	MK312605
II	650	590	310 + 310	570 + 80	*Nakazawaea holstii*	2	MK312606
III	640	320 + 240 + 80	340 + 225 + 75	425 + 215	*Rhodotorula mucilaginosa*	3	MK312607
IV	650	300 + 300 + 50	325 + 325	420 + 150 + 90	*Debaryomyces hansenii*	4	MK312608
V	370	210 + 180	180 + 160	370	*Clavispora lusitaniae*	7	MK312609
VI	625	300 + 265 + 60	320 + 300	400 + 115 + 90	*Meyerozyma guilliermondii*	8	MK312610
VII	450	250 + 200	230 + 220	450	*Candida sake*	9	MK312611
VIII	640	320 + 240 + 80	340 + 225 + 75	425 + 215	*Rhodotorula mucilaginosa*	11	MK312612
IX	640	320 + 240 + 80	340 + 225 + 75	425 + 215	*Rhodotorula mucilaginosa*	14	MK312613
X	370	210 + 180	180 + 160	370	*Clavispora lusitaniae*	15	MK312614
XI	370	210 + 180	180 + 160	370	*Clavispora lusitaniae*	16	MK312615
XII	650	300 + 300 + 50	325 + 325	420 + 150 + 90	*Debaryomyces hansenii*	17	MK312616
XIII	370	210 + 180	180 + 160	370	*Clavispora lusitaniae*	18	MK312617
XIV	450	170 + 100 + 100 + 80	250 + 200	340 + 80 + 30	*Pichia fermentans*	21	MK312618
XV	650	300 + 300 + 50	325 + 325	420 + 150 + 90	*Debaryomyces hansenii*	22	MK312619
XVI	650	300 + 300 + 50	325 + 325	420 + 150 + 90	*Debaryomyces hansenii*	23	MK312620
XVII	500	300 + 100	300 + 150	500	*Tausonia pullulans*	25	MK312621

**Table 2 ijms-21-09699-t002:** Growth of sauerkraut isolates on substrates containing different concentrations of NaCl, lactic acid or different pH and their killer activity.

Isolate	Control (pH 5.6)	NaCl				Lactic Acid			pH			Killer Activity [mm]
		5%	6%	8%	10%	6 g L^−1^	8 g L^−1^	10 g L^−1^	3.6	3.4	3.2	
*Cryptococcus macerans*												
I	+ + + +	+ + + +	+ + +	+ + +	+	+ +	+	-	+ +	-	-	16
*Nakazaweae holstii*												
II	+ + + +	+ +	-	-	-	-	-	-	+ +	-	-	-
*Rhodotorula mucilaginosa*												
III	+ + + +	+ + + +	+ + +	+ + +	-	+ +	+ +	-	+	-	-	-
VIII	+ + + +	+ + +	+ + +	+ +	+	+ +	+	-	+ + +	+	-	-
IX	+ + + +	+ + + +	+ + +	+ + +	+ +	+	+	-	+	-	-	-
*Debaryomyces hansenii*												
IV	+ + + +	+ + +	+ + +	+ + +	+ +	+ + +	+ + +	+ +	+ +	-	-	16
XII	+ + + +	+ + + +	+ + + +	+ + +	+ +	+ +	-	-	+	-	-	24
XV	+ + + +	+ + + +	+ + + +	+ + + +	+ + +	+	+	-	+	+	-	14
XVI	+ + + +	+	-	-	-	+	-	-	+	-	-	11
*Clavispora lusitaniae*												
V	+ + + +	+ + + +	+ + +	+ + +	+ +	+ + +	+ +	-	+ + +	+	-	22
X	+ + + +	+	-	-	-	+ +	+	-	+	-	-	16
XI	+ + + +	+ + + +	+ + +	+ +	+	+ +	+	-	+ +	+	-	6
XIII	+ + + +	+ + + +	+ + +	+ +	+ +	+ +	+	-	+ +	-	-	-
*Meyerozyma guilliermondii*												
VI	+ + + +	+ + + +	+ + +	+ + +	+ + +	+ +	+ +	-	+ + +	+ +	-	-
*Candida sake*												
VII	+ + + +	+ + + +	+ + +	+ +	+	+ +	+ +	-	+ + +	+	-	22
*Pichia fermentans*												
XIV	+ + + +	+	-	-	-	+	+	-	+	+	-	18

(+ + + +) growth diameter over 6 mm, (+ + +) growth diameter 4–6 mm, (+ +) growth diameter 2–4 mm, (+) growth diameter over less than 2 mm, (-) no growth or inhibition detected.

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
