# Peer review of "Yeast Microbiota during Sauerkraut Fermentation and Its Characteristics"

_ijms, 2020, doi:10.3390/ijms21249699_

Round 1

Reviewer 1 Report

    1. Line 183 - 185 - in the presented article changes in the number of yeast during the storage of sauerkraut were not investigated and according to the results (line 82) this number decreased in the following days of fermentation, reaching "0" on the 7th day. So can such a statement be used „In aerobic or semi-anaerobic conditions, the amount of 184 yeast cells is maintained at a relatively high level throughout the fermentation period and may also 185 increase during storage of the finished product”?                                                                         
    1. Line 305, 313, 368 and 376 - please explain why the different temperature was used - 20 ° C during fermentation, 25 ° C during isolation and again 20 ° C when testing the selective stress factors and killer activity?
    2. Line 353and 358 - please standardize the spelling of SPME-GC-MSTOF or SPME-GC-TOFMS

Author Response

Reviewer 1

  1. Line 183 - 185 - in the presented article changes in the number of yeast during the storage of sauerkraut were not investigated and according to the results (line 82) this number decreased in the following days of fermentation, reaching "0" on the 7th day. So can such a statement be used „In aerobic or semi-anaerobic conditions, the amount of 184 yeast cells is maintained at a relatively high level throughout the fermentation period and may also 185 increase during storage of the finished product”?

The sentence was deleted.

  1. Line 305, 313, 368 and 376 - please explain why the different temperature was used - 20 ° C during fermentation, 25 ° C during isolation and again 20 ° C when testing the selective stress factors and killer activity?

Thank You for the remark of the reviewer. The temperaturÄ™ during isolation was changed into 20 ° C. It was our mistake during preparation of the manuscript.

  1. Line 353and 358 - please standardize the spelling of SPME-GC-MSTOF or SPME-GC-TOFMS

Corrected.

Reviewer 2 Report

The manuscript presents a study regarding the biodiversity of yeasts present during fermentation of different varieties of cabbages, as well as a characterization of the obtained yeast isolates.

The title is reflecting the content of article. The introduction provides sufficient background about the factors that influencing the traditional fermentation process and the quality of the sauerkraut. Also, the introduction includes relevant references.

The manuscript does not highlight the novelty of the work.

The aim is clearly pointed in the paper. The originality and scientific reliability of the work are good.   

The research design is appropriate to determine biodiversity of yeasts present during fermentation of cabbages, as well as to characterize the yeast isolates. The research undertaken is adequately described and the materials, methods and conditions are well defined, offering sufficient details for replicability.

The work is indeed carried out in a careful manner. The methods are adequately described. All the figures and tables are relevant for the work at hand. The work presentation is coherent and well described.

The results presented are relevant to the aim of the study. The interpretation is warranted by the data and support the conclusion of the work that during properly conducted fermentation process of sauerkraut, with anaerobic conditions, yeast growth occurs at all times and is present in the initial period of fermentation.

References are up to date and relevant. The most current literature has been used.

Author Response

Regarding the remark - The manuscript does not highlight the novelty of the work.

The novelty of the work was exposed in the following fragments of the discussion:

l.177-178 – „Very little attention was paid to the presence of yeast during this process, and available publications come from many years ago, or contain only mention of yeasts as spoilage organisms [6,8].”

l.219-220 – „This is the first report regarding the presence of these yeast {Clavispora lusitaniae} in fermented vegetable.”

  1. 225-229 – ” This microorganism is most often found with D. hansenii and lactic acid bacteria, but there is no references on the properties of cultures, as well as resistance to stress factors occurring during sauerkraut fermentation, such as the presence of NaCl, lactic acid, low pH, etc. Our research is also the first to determine the ability of these yeasts to show killer activity and produce volatile compounds.” – regarding Clavispora lusitaniae

We added also to the Introduction section the following fragment – „Most of the so far published articles related to the microbiology of the sauerkraut concern LAB. There is very little information regarding the presence of yeast during fermentation of sauerkraut and their characteristics”.